# Plant-Associated Novel Didymellaceous Taxa in the South China Botanical Garden (Guangzhou, China)

**DOI:** 10.3390/jof9020182

**Published:** 2023-01-29

**Authors:** Nuwan D. Kularathnage, Indunil C. Senanayake, Dhanushka N. Wanasinghe, Mingkwan Doilom, Steven L. Stephenson, Jiage Song, Wei Dong, Biao Xu

**Affiliations:** 1Innovative Institute for Plant Health/Key Laboratory of Green Prevention and Control on Fruits and Vegetables in South China, Ministry of Agriculture and Rural Affairs, Zhongkai University of Agriculture and Engineering, Guangzhou 510225, Guangdong, China; 2Center of Excellence in Fungal Research, Mae Fah Luang University, Chiang Rai 57100, Thailand; 3School of Science, Mae Fah Luang University, Chiang Rai 57100, Thailand; 4Center for Mountain Futures, Kunming Institute of Botany, Honghe County 654400, Yunnan, China; 5Department of Biological Sciences, University of Arkansas, Fayetteville, AR 72701, USA

**Keywords:** 3 new species, 1 new host record, Dothideomycetes, Guangdong Province, hyaline-spored coelomycetes, Pleosporales

## Abstract

The South China Botanical Garden (SCBG), one of the largest and oldest botanical gardens in China, conserves important plant germplasms of endangered species. Therefore, ensuring tree health and studying the associated mycobiome of the phyllosphere is essential to maintaining its visual aesthetics. During a survey of plant-associated microfungal species in SCBG, we collected several coelomycetous taxa. Phylogenetic relationships were evaluated based on the analyses of ITS, LSU, RPB2, and *β*-*tubulin* loci. The morphological features of the new collections were compared with those of existing species, emphasizing close phylogenetic affinities. Based on the morphological comparisons and multi-locus phylogeny, we introduce three new species. These are *Ectophoma phoenicis* sp. nov., *Remotididymella fici*-*microcarpae* sp. nov., and *Stagonosporopsis pedicularis*-*striatae* sp. nov. In addition, we describe a new host record for *Allophoma tropica* in the *Didymellaceae*. Detailed descriptions and illustrations are provided along with notes comparing allied species.

## 1. Introduction

The South China Botanical Garden (SCBG), Chinese Academy of Sciences, Guangzhou was established in 1929 [1]. This is the largest modern botanical garden in Guangzhou, Guangdong Province, and it extends over 2854 acres comprising around 2400 plant species including alpine, arctic, aquatic, Mediterranean, tropical, and desert examples [1]. SCBG is one of the top plant germplasm conservation institutions in China and it contains 710 of the verified Red List plant taxa [2]. There are more than 14,000 trees including Chinese endemic species in ex-situ conservation [3]. Tree health and disease management are rather important in SCBG.

*Didymellaceae* was established by de Gruyter et al. [4] to accommodate *Phoma* and other allied genera. Aveskamp et al. [5] delimited the boundaries of the *Didymellaceae* based on morphology and combined ITS, LSU, SSU, and *βtubulin* loci analyses. Subsequent revisions of the *Didymellaceae* resolved generic boundaries and revealed more natural evolutionary relationships [6,7,8,9,10,11]. Currently, the family encompasses more than 5400 species in 44 accepted genera [12,13,14] (Species Fungorum. http://www.speciesfungorum.org/Names/Names.asp, accessed on 14 August 2022). Species of the *Didymellaceae* are cosmopolitan and distributed in a wide range of hosts and habitats [5,15,16]. Some species are plant pathogens causing leaf and stem lesions and fruit rots [5,7,15,17], while some species such as *Allophoma hayatii* Babaahm. and Mehr-Koushk., *Calophoma petasitis* Tibpromma, Camporesi, and K.D. Hyde are saprobes on dead plant parts [18,19,20]. Few species such as *Phoma herbarum* Westend. and *Juxtiphoma eupyrena* (Sacc.) Valenz-Lopez, Crous, Stchigel, and Guarro and Cano have been reported as human and animal pathogens [21,22].

In a survey of fungal species associated with plants in botanical gardens, we collected several saprobic, hyaline-spored, and didymellaceae-like coelomycetous taxa from the SCBG. The main objectives of this study are to analyze the taxonomic placement of our didymellaceous collections and then to describe the taxonomic novelties using morphology and the multi-locus phylogeny of ITS, LSU, RPB2, and *β*-*tubulin* sequences. Our study revealed that three of our isolates are new to science. Herein, we introduce these isolates as novel species namely *Ectophoma phoenicis* sp. nov., *Remotididymella fici*-*microcarpae* sp. nov., and *Stagonosporopsis pedicularis*-*striatae* sp. nov. A new collection of *Allophoma tropica* (R. Schneid. and Boerema) Qian Chen and L. Cai is described herein as a new host record. Detailed descriptions, illustrations, and notes are provided. 

## 2. Material and Methods

### 2.1. Sampling Sites, Specimens, and Isolates

Dead plant specimens were collected from the south China Botanical Garden, Guangzhou, Guangdong Province, China (Figure 1), between June to September 2021. Specimens were detached from the host using sterile blades and packed in sterilized paper bags. The collection site is characterized by a tropical climate with abundant sunshine and rainfall throughout the year. The average annual temperature is 22 °C with around 2125 mm of rainfall per year [23].

The collected dead samples (petioles, sepals, and stems) were brought to the laboratory in sterilized paper bags and examined with a stereomicroscope (Carl Zeiss Discovery V8). Conidia were cultured following the method described by Senanayake et al. [24]. The germinated conidia were aseptically transferred into fresh potato dextrose agar (PDA) plates, incubated at 25 °C in the dark to obtain pure cultures, and later transferred to PDA slants and stored at 4 °C for further study. Colony characters were recorded from PDA cultures. Fungarium specimens are deposited at the Herbarium of Zhongkai University of Agriculture and Engineering (ZHKU), and all the ex-type and living cultures are deposited at the Culture Collection of Zhongkai University of Agriculture and Engineering (ZHKUCC). Index Fungorum numbers (https://www.indexfungorum.org, accessed on 1 June 2022) and Facesoffungi numbers [25] were registered for the new species.

### 2.2. Morphological Studies

Microscopic mounts of fruiting structures in sterilized tap water were examined and photographed using a stereomicroscope fitted with a camera (ZEISS Axiocam 208). The micromorphological characteristics such as the structure of the conidiomatal wall, conidiogenous cells, and conidia were studied and photographed using a compound microscope (Nikon Eclipse 80i) fitted with a digital camera (Canon 450D). All microscopic measurements were made with the Tarosoft image framework (v. 0.9.0.7).

### 2.3. DNA Extraction, PCR Amplification, and Sequencing

Fresh mycelia grown on PDA for two weeks at 25 °C in the dark were used for DNA extraction using a fungal genomic DNA extraction kit (Biospin DNA Extraction Kit, Bioer Technology, Co. Ltd., Hangzhou, China) following the manufacturer’s protocols. Polymerase chain reactions (PCR) and sequencing were carried out for the complete ITS region, part of the LSU ribosomal DNA, the RNA polymerase II subunit (RPB2) gene, and the *β*-*tubulin* gene. The PCR amplification reactions were carried out with the following protocols in Table 1. The total volume of the PCR reaction was 25 µL containing 1 µL of DNA template, 1 µL of each forward and reverse primer, 12.5 µL of 2 × PCR Master Mix, and 9.5 µL of double-distilled sterilized water (ddH_2_O). All the PCR thermal cycles include a final hold at 4 °C.

The PCR products were observed on a 1% agarose electrophoresis gel stained with ethidium bromide. Purification and sequencing of PCR products were carried out at Sunbiotech Company, Beijing, China. Sequence quality was checked, and sequences were condensed with DNASTAR Lasergene v. 7.1 [26]. Sequences derived in this study were deposited in GenBank and accession numbers were obtained (Table 2).

### 2.4. Phylogenetic Analyses

BLASTn searches were made using NCBI BLASTn (https://blast.ncbi.nlm.nih.gov/Blast.cgi, accessed on 18 March 2022). The newly generated sequences assist in taxon sampling for phylogenetic analyses. All the ex-type strains of species were included if available, and other authentic strains were selected when sequences from ex-type strains were unavailable. [27,28,29,30,31,32,33] were followed to obtain sequences from GenBank (Table 2). The concatenated ITS, LSU, RPB2, and *β*-*tubulin* sequence dataset for the family *Didymellaceae* comprised 123 strains with the outgroup taxon as *Leptosphaeria conoidea* (De Not.) Sacc. (CBS 616.75) and *Leptosphaeria doliolum* (Pers.) Ces. & De Not. (CBS 505.75). DNA sequences of all the single gene regions were aligned using the online version of MAFFT v. 7.0362 [34] with default settings and manually adjusted using BioEdit 7.1.3 [35] when necessary.

Maximum likelihood analysis was performed by RAxML [36] implemented in raxmlGUIv. 1.5 [37] using the ML + rapid bootstrap setting and the GTR + I + G model of nucleotide substitution with 1000 replicates. For the Bayesian inference (BI) analyses, the optimal substitution model for the combined datasets was determined to be GTR + I + G using the MrModeltest software v. 2.2 [38]. The BI analyses were computed in MrBayes v. 3.2.6 [39] with four simultaneous Markov chain Monte Carlo chains from random trees over 5 M generations (trees were sampled every 1000th generation). The distribution of log-likelihood scores was observed to check whether sampling was in the stationary phase and Tracer v1.5 was used to check if further runs were required to reach convergence [40]. The consensus tree and posterior probabilities were calculated after discarding the first 20% of the sampled trees as burn-in. The phylogram was visualized in FigTree v. 1.4 [41].

### 2.5. PHI Analyses

The PHI test was performed using SplitsTree4 v. 4.17.1 to determine the recombination level within phylogenetically closely related species. The concatenated four-locus dataset (ITS + LSU + RPB2 + *β*-*tubulin*) was used for the analyses. PHI test results (Φw) ≥0.05 indicated no significant recombination within the dataset. The relationships between closely related taxa were visualized in split graphs with both the Log-Det transformation and splits decomposition options.

## 3. Results

### 3.1. Phylogenetic Analyses

The alignment comprised 2308 nucleotide characters (488 of ITS, 893 of LSU, 595 of RPB2, 332 of *β*-*tubulin*). Maximum likelihood analysis yielded the best ML tree (Figure 2) with a likelihood value of −25,904.846543 and the following model parameters: estimated base frequencies A = 0.246664, C = 0.258093, G = 0.262080, and T = 0.233164; substitution rates AC = 1.535206, AG = 4.889606, AT = 1.722258, CG = 0.744297, CT = 10.883910, and GT = 1.0; proportion of invariable sites I = 0.587394; gamma distribution shape parameter: α = 0.536998. The alignment contained a total of 857 distinct alignment patterns and 9.44% of undetermined characters.

After discarding the first 20% of generations in the Bayesian analyses, 4000 trees remained from which the 50% consensus tree and Bayesian Interference (BI) posterior probabilities were calculated (Figure 2). All individual trees generated under different criteria from single gene datasets were similar in topology and not significantly different from the final trees generated from the concatenated datasets of the *Didymellaceae*. Topologies of the ML and Bayesian trees were similar to each other and there were no significant differences.

The phylogenetic analyses of this study (Figure 2) showed that the *Didymellaceae* comprises a total of 27 well-supported genera. We included 32 sequences from four new collections representing eight isolates in this analysis. Newly generated sequences from two isolates (ZHKUCC 22-0167, ZHKUCC 22-0168) grouped with *Stagonosporopsis astragali* (Cooke and Harkn.) Aveskamp, and Gruyter and Verkley (CBS 178.25) with strong statistical support (89% in ML, 0.90 in BI). Furthermore, another two isolates (ZHKUCC 22-0169, ZHKUCC 22-0170) grouped with the type strain (CBS 537.66) and another strain (CBS 436.75) of *Allophoma tropica* with a strong support value (99% in ML, 0.99 in BI). Other two isolates (ZHKUCC 22-0163, ZHKUCC 22-0164) form a distinct basal clade to *Ectophoma multirostrata* (P.N. Mathur, S.K. Menon and Thirum.) Valenz.-Lopez, Cano, Crous, Guarro and Stchigel-*E*. *Iranica* M. Mehrabi, Larki and Farokhinejad subclade with strong statistical support (98% in ML, 0.99 in BI). Moreover, two new isolates (ZHKUCC 22-0165, ZHKUCC 22-0166) form a sister clade to *Remotididymella ageratinae* H.B. Zhang, A.L. Yang and L. Chen (G1338) with strong support value (100% in ML, 1.00 in BI).

### 3.2. Taxonomy

***Allophoma*** Q. Chen & L. Cai, Stud. Mycol. 82: 162 (2015).

Type species: *Allophoma tropica* (R. Schneid. and Boerema) Q. Chen and L. Cai. Chen et al.

**Notes:** *Allophoma* was introduced to accommodate phoma-like taxa with various-shaped conidia [7]. Currently, 14 species are accepted in the genus [14] (Species Fungorum. http://www.speciesfungorum.org/Names/Names.asp, accessed on 14 August 2022). *Allophoma* species are phytopathogens or saprobes on leaves, pods, and stems and some are human and animal pathogens [33,42,43]. *Allophoma labilis* (Sacc.) Qian Chen and L. Cai often causes leaf necrosis, canker, and stem lesions, or stem rot in various plants [18,44,45,46,47,48], and *A*. *tropica* can cause necrosis in a wide variety of ornamental plants [42]. In addition, *A. cylindrispora* Valenz-Lopez, Stchigel, Guarro and Cano was isolated from a lesion of the human eye [43] and *A*. *oligotrophica* Qian Chen, Crous and L. Cai is an air borne fungus [49].

***Allophoma* *tropica*** (R. Schneid. and Boerema) Q. Chen and L. Cai, Stud. Mycol. 82: 164 (2015).

Basionym: *Phoma tropica* R. Schneid. and Boerema, Phytopath. Z. 83 (4): 361 (1975)

Index Fungorum number: IF814071; Facesoffungi number: FoF 13240, Figure 3.

*Saprobic* on the sepals of *Canna* sp. **Sexual morph**: Undetermined. **Asexual morph**: Coelomycetous. Appearing as black spots. *Hyphae* 4–6 μm wide, initially hyaline, become olivaceous to brown, septate. *Conidiomata* 80–250 μm high, 100–300 μm diam. (x = 210 × 280 μm, *n* = 20), pycnidial, mostly solitary, rarely aggregated, superficial to erumpent, uniloculate, subglobose to irregular, dark brown, glabrous, conspicuous dark ostiole. *Ostiole* single, dark, distinct, non-papillate or sometimes slightly papillate. *Pycnidial wall* 5–10 μm thick (x = 8 μm, *n* = 10), 1–3-layered, composed of pale brown cells of *textura angularis*. *Conidiophores* are reduced to conidiogenous cells. *Conidiogenous cells* 5–8 × 3–4 μm (x = 7 × 3.6 μm, *n* = 10), phialidic, ampulliform to doliiform, hyaline, smooth. *Conidia* 3–5 × 1–2.5 μm (x = 4 × 2 μm, *n* = 20), ellipsoidal, smooth and thin-walled, hyaline, aseptate, with small guttules. Conidial exudates not observed. *Chlamydospores* globose 10–25 μm diam. (x = 20 μm, *n* = 20) to irregular 10–15 × 6–15 μm (x = 13 × 10 μm, *n* =20), unicellular or multicellular, intercalary or terminal, solitary or in chains, smooth, verruculose or incidentally tuberculate, subhyaline to pale brown.

**Culture characters:** Colonies on PDA reaching 5 cm after 7 days at 25 °C in the dark, white to off-white, aerial mycelium less, circular, flat, smooth to slightly waved margin; reverse off-white. Cultures were not sporulating and no pigments were produced.

**Material examined:** China, Guangdong Province, Guangzhou City, South China Botanical Garden (23°11′12″ N 113°21′51″ E), on sepals of *Canna* sp. (*Cannaceae*), 17 June 2021, N.D. Kularathnage, NDK 71-3 (ZHKU 22-0099), living cultures ZHKUCC 22-0169, ZHKUCC 22-0170.

**Hosts and distribution:** on *Aphelandra* sp. from the Netherlands, on *Saintpaulia ionantha* from Germany, on *Gossypium* sp. from Bolivia [42], on *Lactuca sativa* from Italy [50], on *Syzygium cumini* from India [46], and on sepals of *Canna* sp. from China (the present study).

**Notes**: The multi-locus phylogeny of ITS, LSU, RPB2, and *β*-*tubulin* showed that our isolates (ZHKUCC 22-0169 and ZHKUCC 22-0170) are affiliated with *Allophoma tropica* (CBS 537.66, CBS 436.75) with high statistical support (ML/BI 99/0.99) (Figure 2). A single gene comparison of ITS, LSU, RPB2, and *β*-*tubulin* of our isolates (ZHKUCC 22-0169, ZHKUCC 22-0170) with the type strain of *Allophoma tropica* (CBS 436.75) revealed base pair differences of 2/488, 1/893, 0/595, and 2/332, respectively. Our specimen fits well with the type collection of *Allophoma tropica* (DSM 63365) [51] in all morphological aspects [52].

No species of *Allophoma* had been reported from *Canna* sp., and our collection is the first *Allophoma* species reported from this host genus. Two species, *A*. *pterospermicola* Qian Chen and L. Cai and *A*. *thunbergiae* Jun Yuan and Yong Wang, have been described from Guangxi and Guizhou Provinces in China [49,53], and *A*. *tropica* is the third species reported in the country.

***Ectophoma*** Valenz-Lopez, Cano, Crous, Guarro and Stchigel, Stud. Mycol. 90: 34 (2017)

*Type species: Ectophoma multirostrata* (P.N. Mathur, S.K. Menon and Thirum.) Valenz-Lopez, J.F. Cano, Crous, Guarro, and Stchigel.

**Notes:** *Ectophoma* was established to accommodate two phoma-like taxa, *Phoma multirostrata* (P.N. Mathur, S.K. Menon and Thirum.) Dorenb. and Boerema and *P. pereupyrena* Gruyter, Noordel. and Boerema which formed a distinct lineage in the *Didymellaceae* [28]. Currently, there are four accepted species (Species Fungorum. http://www.speciesfungorum.org/Names/Names.asp, accessed on 14 August 2022). This genus is characterized by pycnidial conidiomata with one or more short necks, phialidic conidiogenous cells, and aseptate conidia [28]. *Ectophoma* species have been isolated from different woody and herbaceous plants as opportunistic plant pathogens or from soil inhabitants [51,54]. *Ectophoma* species have been reported from Greece, India, Iran, South Africa, and The Netherlands.

***Ectophoma phoenicis*** Kular. sp. nov.

Index Fungorum number: IF900158; Facesoffungi number: FoF 13241; Figure 4.

Etymology: in reference to the host genus, *Phoenix*.

*Saprobic* on dead petioles of *Phoenix roebelenii* O’Brien. **Sexual morph**: Undetermined. **Asexual morph**: Coelomycetous, *Conidiomata* 80–110 μm high, 100–130 μm wide (x = 100 × 120 μm, *n* = 10), pycnidial, solitary or rarely aggregated, immersed in substrate, conical to subglobose, brown to blackish brown, coriaceous, ostiolate, apapillate. *Pycnidial wall* comprises 4–7 layered, brown cells of *textura angularis*. *Conidiophores* are reduced to conidiogenous cells. *Conidiogenous cells* 15–20 μm high, 3–8 μm wide (x = 18 × 5 μm, *n* = 10), phialidic, cylindrical to doliiform, discrete, hyaline to grey, smooth. *Conidia* 6–8 × 2–4 μm (x = 7 × 3 μm, *n* = 20), oblong to ellipsoidal, aseptate, with polar guttules, hyaline, smooth, thin-walled, straight.

**Culture characters:** Colonies on PDA reaching 4 cm diam. after 7 days of incubation at 25 °C, circular, flat, floccose, filiform, pale brown to greyish brown, darker in the central area, then paler ring and darker ring around the center; reverse brown. Hyphae pale brown, branched, septate. No pigments or chlamydospores observed.

**Material examined:** China, Guangdong Province, Guangzhou City, South China Botanical Garden (23°11′12″ N 113°21′51″ E) on dead petioles of *Phoenix roebelenii* (*Arecaceae*), 17 June 2021, N.D. Kularathnage, NDK 78-1 (ZHKU 22-0093, **holotype**), ex-type culture ZHKUCC 22-0163; *ibid*. NDK 78-2 (ZHKU 22-0094, isotype), ex-type culture ZHKUCC 22-0164.

**Notes:** The multi-locus analysis of ITS, LSU, RPB2, and *β*-*tubulin* showed that our isolates (ZHKUCC 22-0163, ZHKUCC 22-0164) clustered within *Ectophoma* and formed a sister clade to *E*. *iranica* and *E*. *multirostrata* with 98% in ML and 0.99 in BI support values (Figure 2). A single gene comparison of ITS, RPB2, and *β*-*tubulin* locus of our isolates (ZHKUCC 22-0163, ZHKUCC 22-0164) with the type strain of *E*.*iranica* (CBS 144681) and *E*. *multirostrata* (CBS 274.60) revealed the base pair differences of 7/488, N/A, 3/332 and 6/488, 6/595, 2/332, respectively. The LSU sequences of all *Ectophoma* species are identical. The genetic distinctness and phylogenetic stability of our isolates were further confirmed by PHI analysis of the *Ectophoma* clade. The result showed that Φw = 0.99 and this means there was no significant genetic recombination (Φw ≥ 0.05) between these novel isolates with existing *Ectophoma* species (Figure 5).

Morphologically, our collection differs from *E*. *iranica* by its floccose, grey colonies, apapillate pycnidia, cylindrical to doliiform, hyaline to grey conidiogenous cells, and ellipsoidal, straight conidia. In contrast, *E*. *iranica* has pale brown to greyish brown colonies, conidiomata with 1–2(3)-narrowed necks, and hyaline to pale brown, oblong to ellipsoidal, straight or sometimes very slightly curved conidia [29]. *Ectophoma iranica* is a phytopathogen that forms leaf spots on *Dracaena compacta* and *Catharanthus roseus* [29] while our collections are saprobes. Furthermore, our collections differ from *E*. *multirostrata* by its bi-guttulated conidia, and conidiomata with single ostiole while *E*. *multirostrata* is characterized by eguttulate conidia or sometimes with 2–3, polar guttules and necks with several ostioles. *Ectophoma multirostrata* is a plurivorous opportunistic plant pathogen isolated from soil and also from some plant samples [51] and our species is a saprobe.

***Remotididymella*** Valenz-Lopez, Crous, Cano, Guarro and Stchigel, Stud. Mycol. 90: 35 (2017).

Type species: *Remotididymella destructiva* (Plowr.) ValenzuelaLopez, Cano, Crous, Guarro and Stchigel.

**Notes:***Remotididymella* is characterized by aseptate, hyaline, smooth- and thin-walled, allantoid or cylindrical, guttulate conidia. Currently, there are eight species listed under this genus (https://www.indexfungorum.org, accessed on 1 June 2022) and the sexual morph has been reported only for *R*. *bauhiniae* Jayasiri, E.B.G. Jones and K.D. Hyde [48]. Species of *Remotididymella* have been reported from *Ageratina adenophora*, *Bauhinia* sp., *Capsicum annuum*, *Lycopersicon* sp., and *Solanum* sp. as saprobes or pathogens. However, some species have been isolated from air, soil in tropical forests, and human respiratory tract [12,28,32,48]. *Remotididymella* species have been reported from China, Guadeloupe, Papua New Guinea, Thailand, The Republic of Fiji, and the United States.

***Remotididymella fici-microcarpae*** Kular. sp. nov.

Index Fungorum number: IF900161; Facesoffungi number: FoF 13242; Figure 6.

Etymology: in reference to the host name *Ficus microcarpa*.

*Saprobic* on dead stems of *Ficus microcarpa* L.f **Sexual morph**: Undetermined. **Asexual morph:**
*Conidiomata* 180–230 × 130–180 μm, (x = 200 × 150 μm, *n* = 20), pycnidial, dark brown, mostly solitary, rarely aggregated, immersed, glabrous, conical to irregularly-shaped, with a single papillate ostiolar neck. *Pycnidial wall* 15–25 μm thick (x = 19 μm, *n* = 10), 6–8-layered, composed of brown, flattened cells of *textura angularis*. *Conidiogenous cells* 12–15 × 8–11 μm (x = 14 × 9 μm, *n* = 20), phialidic, ampulliform, hyaline, smooth-walled. *Conidia* 5–8 × 3.5–4.5 μm (x = 7 × 4 μm, *n* = 20), mostly fusiform to allantoid, aseptate, hyaline, smooth, thin-walled, indistinct guttules.

**Culture characters:** Colonies on PDA reached 6 cm diam. after 7 days, circular, flat, filiform margin, center pale, margin darker, margin comprised of filiform hyphal tips, aerial mycelia less, brown, reverse pale brown. Cultures not sporulating and no pigments were produced.

**Material examined:** China, Guangdong Province, Guangzhou City, South China Botanical Garden (23°11′12″ N 113°21′51″ E), on stems of *Ficus microcarpa* (*Moraceae*), 17 June 2021, Senanayake I.C, S2 3-1 (ZHKU 22-0095, **holotype**), ex-type culture ZHKUCC 22-0165, *ibid*. S2 3-1 A (ZHKU 22-0096, isotype), ex-type living culture ZHKUCC 22-0166.

**Notes:** The combined gene analysis of ITS, LSU, RPB2, and *β-tubulin* (Figure 2) showed that our isolates (ZHKUCC 22-0165, ZHKUCC 22-0166) grouped with the type strain of *Remotididymella ageratinae* (CGMCC 3.19991) with a strong support value (100% in ML, 1.00 in BI). Result of PHI analysis of *Remotididymella* species showed a value of Φw = 0.26 and there was no significant genetic recombination (Φw ≥ 0.05) between these novel species of *Remotididymella* and other species in this genus (Figure 7). A comparison of the DNA sequences of ITS, LSU, RPB2, and *β*-*tubulin* locus of our isolates (ZHKUCC 22-0165, ZHKUCC 22-0166) with the type strain of *R*. *ageratinae* revealed base pair differences of 6/488, 8/893, 9/595, 3/332, respectively.

Morphologically, our collections differ from *R*. *ageratinae* by having small conidiomata (180–230 × 130–180 μm), large conidiogenous cells (12–15 × 8–11 μm), and fusiform to allantoid conidia with indistinct guttules, whereas *R*. *ageratinae* has large conidiomata (107–409 × 121–503 µm), small conidiogenous cells (8–9 × 10 µm), and oblong to cylindrical, obovoid, sometimes slightly curved to reniform conidia with distinct guttules. Therefore, we introduce our collection as *Remotididymella fici*-*microcarpae* sp. nov. based on morphology and phylogeny.

***Stagonosporopsis*** Died., Annls mycol. 10 (2): 142 (1912).

Type species: *Stagonosporopsis hortensis* (Sacc. and Malbr.) Petr.

**Notes:** There are 52 accepted species in this genus (Species Fungorum. http://www.speciesfungorum.org/Names/Names.asp, accessed on 14 August 2022). Members of the *Stagonosporopsis* are saprobes on dead plant materials, and some species have been reported from house dust and garden soil [12]. Some species of *Stagonosporopsis* can cause devastating diseases on a wide range of economically important plants, including those found in farmlands, forests, grasslands, and other natural ecosystems [5]. *Stagonosporopsis* species have been reported as severe pathogens of some crops and ornamentals in several countries, including Australia China, France, India, Italy, Turkey, and the United States [55,56,57,58,59].

***Stagonosporopsis pedicularis-striatae*** Kular. sp. nov.

Index Fungorum number: IF900160; Facesoffungi number: FoF 13243, Figure 8.

Etymology: in reference to the host name *Pedicularis striata.*

*Saprobic* on dead stems of *Pedicularis striata* Pallas. **Sexual morph:** Undetermined. **Asexual morph:** Coelomycetous. *Conidiomata* 150–200 × 300–400 μm (x = 180 × 350 μm, *n* = 10), pycnidial, solitary or aggregated, scattered, subglobose, coriaceous, brown to dark brown, thin-walled, glabrous, ostiolate. *Ostiole* single, slightly papillate. *Pycnidial wall* 10–15 μm thick, pseudoparenchymatous, 4–5-layered, composed with brown, *angularis* to *globosa* cells. *Conidiophores* arereduced to conidiogenous cells. *Conidiogenous cells* 4–6 × 4–5 μm (x = 5.5 × 4.5 μm, *n* = 20), phialidic, subglobose, hyaline, smooth. *Conidia* 7–9 × 3–5 μm (x = 8 × 4 μm, *n* = 20), oblong to ellipsoid, with ends rounded, smooth and thin-walled, aseptate, small guttulate. 

**Culture characters:** Colonies on PDA reached 7 cm diam. after 7 days, circular, flat, smooth, entire margin, aerial mycelia concentrated at the margin, brown, center dark, margin pale; reverse pale brown, center pale, margin darker. Cultures not sporulating and no pigments are produced.

**Material examined:** China, Guangdong Province, Guangzhou City, South China Botanical Garden (23°11′12″ N 113°21′51″ E) on dead petioles of *Pedicularis striata* (*Orobanchaceae*), 17 June 2021, Kularathnage N.D., S1 1010 (ZHKU 22-0097, **holotype**), ex-type culture ZHKUCC 22-0167; *ibid*. S1-1002 (ZHKU 22-0098, isotype), ex-type culture ZHKUCC 22-0168.

**Notes:** Our isolates (ZHKUCC 22-0167, ZHKUCC 22-0168) grouped with *Stagonosporopsis astragali* (CBS 178.25), forming a well-supported (89% in ML, 0.90 in BI) distinct clade in the combined gene analysis of ITS, LSU, RPB2 and *β*-*tubulin* (Figure 2). The result of PHI analysis of *Stagonosporopsis* species in this study shows Φw = 0.18 and there was no significant genetic recombination between these novel isolates and other *Stagonosporopsis* species (Figure 9). Comparison of the DNA sequences of ITS, LSU, RPB2, and *β*-*tubulin* locus of our isolates (ZHKUCC 22-0167, ZHKUCC 22-0168) with the type strain of *S*. *astragali* reveals the base pair differences of 5/488, 9/893, 7/595, 15/332 respectively.

Morphologically, our collections are different from *Stagonosporopsis astragali* by short papillate, larger conidiomata (150–200 × 300–400 μm), and oblong to ellipsoid conidia without polar guttules. In contrast, *S*. *astragali* is an opportunistic pathogen producing smaller conidiomata (80–180 μm diam.) with 0–3 papillate ostioles, and cylindrical to allantoid conidia with numerous polar guttules. Therefore, we introduce this collection as *Stagonosporopsis pedicularis*-*striatae* sp. nov. This fungus has often recorded in North America (United States and Canada) on stems of various species of *Astragalus* [51].

## 4. Discussion and Conclusions

In a survey of the diversity and species richness of plant-associated fungi in the South China Botanical Garden, we collected several saprobic, hyaline-spored asexual species in the *Didymellaceae*. The *Didymellaceae* has recently undergone extensive revision based on its phylogenetic relationships and morphological characteristics [5,7,60]. In this study, we identified and introduced three new species in *Didymellaceae* (*Ectophoma phoenicis*, *Remotididymella fici*-*microcarpae*, and *Stagonosporopsis pedicularis*-*striatae*) along with a new host and locality record of *Allophoma tropica* based on polyphasic approaches according to the procedure of [61,62,63]. The guidelines for introducing novel species have been discussed in [64,65] were followed.

We described and illustrated a new locality report of *Allophoma tropica*. *Allophoma tropica* was recorded for the first time as a saprobe on leaves of *Streptocarpus ionanthus* and this study is the first record of *Allophoma* species on *Canna* in China. Furthermore, this study reported morphological characters of chlamydospores from cultures of *Allophoma* for the first time. The type collection of *Allophoma tropica* obtained from *Streptocarpus ionanthus* (H. Wendl.) Christenh. as a pathogen and morphological characters were obtained from pycnidia in culture while morphological characters of our collection were obtained from conidiomata on the substrate. Therefore, there are negligible differences in the morphological characteristics of fungi on a substrate and in media. There are four *Ectophoma* species listed in (Species Fungorum. http://www.speciesfungorum.org/Names/Names.asp, accessed on 14 August 2022). The type species of *Ectophoma*, *E*. *multirostrata* has been isolated from poultry farm soil and the live stem of *Cucumis sativus* in a greenhouse [28]. *Ectophoma iranica* and *E*. *pomi* are plant pathogens isolated from leaf spots of *Dracaena compacta*, *Catharanthus roseus* [29], and *Coffea arabica* [28]. *Ectophoma insulana* was isolated from the fruit of *Olea europaea* and from house dust [12]. Morphologically, our collections differ from *E*. *iranica*, and *E*. *multirostrata* and also our collection of *Ectophoma phoenicis* isolated from the dead petioles of *Phoenix roebelenii* are saprobic fungi. As such, this is the first record of saprobic behavior of *Ectophoma* species. Our collection of *Ectophoma phoenicis* was identified and introduced as a new species belonging to the genus *Ectophoma*.

*Remotididymella fici*-*microcarpae* is the first species in this genus reported from China and also from *Ficus microcarpa*. This was collected from dead stems as saprobes. However, *Remotididymella* species show a wide variation of life modes including plant and human pathogens and saprobes [12,32,48]. Morphologically, our collections differ from *R*. *ageratinae* by having immersed, brown, small conidiomata, large conidiogenous cells, and fusiform to allantoid conidia with indistinct guttules. Our collection of *Remotididymella fici*-*microcarpae* was identified and introduced as a new species belonging to the genus *Remotididymella*.

There are several *Stagonosporopsis* species that have been reported from China, and most of them are plant pathogens [31]. *Stagonosporopsis vannaccii* Baroncelli, Cafà, Castro, Boufleur and Massola was reported from leaf spots on *Crassocephalum crepidioides* in Guangxi [66] while *Stagonosporopsis cucurbitacearum* (Fr.) Aveskamp, Gruyter and Verkley are the cause of pumpkin gummy stem blight, which is one of the most devastating pumpkin crop diseases in North-East China [67]. *Stagonosporopsis pogostemonis* M. Luo, Y.H. Huang and Manawas. was reported from leaf spots and as a stem blight on *Pogostemon cablin* in South China [31]. However, *Stagonosporopsis pedicularis*-*striatae* was collected from dead stems as a saprobe. Morphologically, our collections differ from *Stagonosporopsis astragali* by their subglobose, slightly papillate, comparatively larger conidiomata (150–200 × 300–400 μm), and oblong to ellipsoid conidia. In terms of phylogenetic and morphological differences, we identified and introduced *Stagonosporopsis pedicularis*-*striatae* as a new species belonging to the genus *Stagonosporopsis.*

## Figures and Tables

**Figure 1 jof-09-00182-f001:**
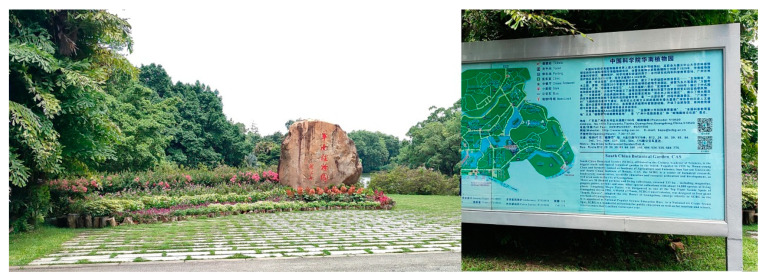
Entrance of Southern China Botanical Garden.

**Figure 2 jof-09-00182-f002:**
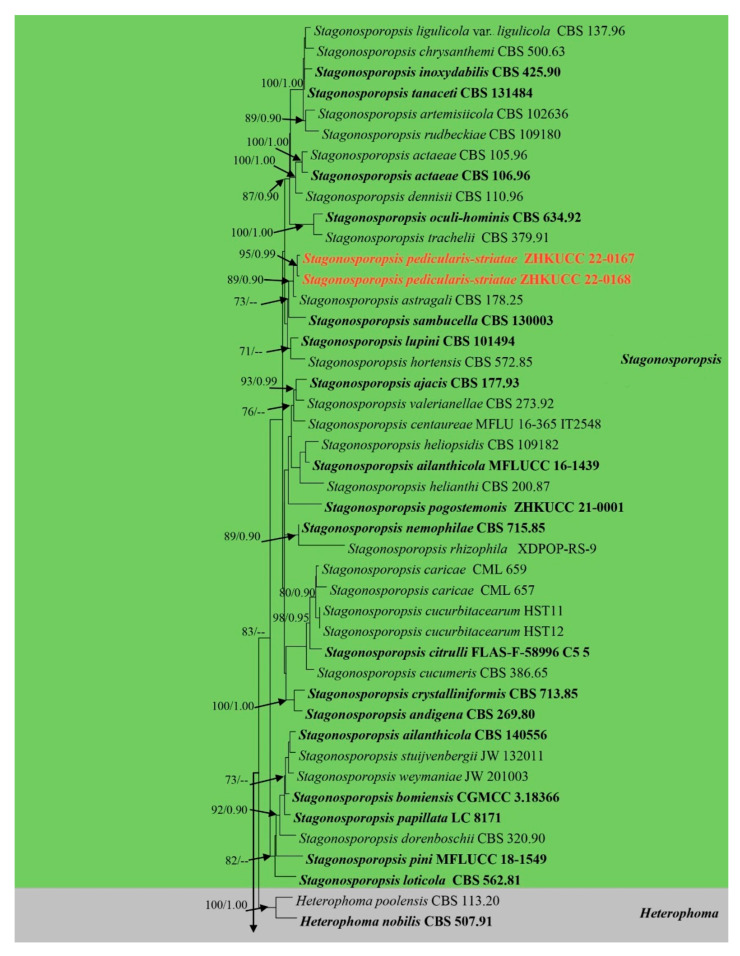
Phylogram generated from a maximum likelihood analysis based on a combined ITS, LSU, RPB2, and *β*-*tubulin* sequence alignment. Maximum likelihood bootstrap support values greater than 70% and Bayesian posterior probabilities greater than 0.90 are given at the nodes. The tree is rooted with *Leptosphaeria conoidea* (CBS 616.75) and *Leptosphaeria doliolum* (CBS 505.75). Ex-type strains are given in bold and the newly generated sequences are indicated in red bold.

**Figure 3 jof-09-00182-f003:**
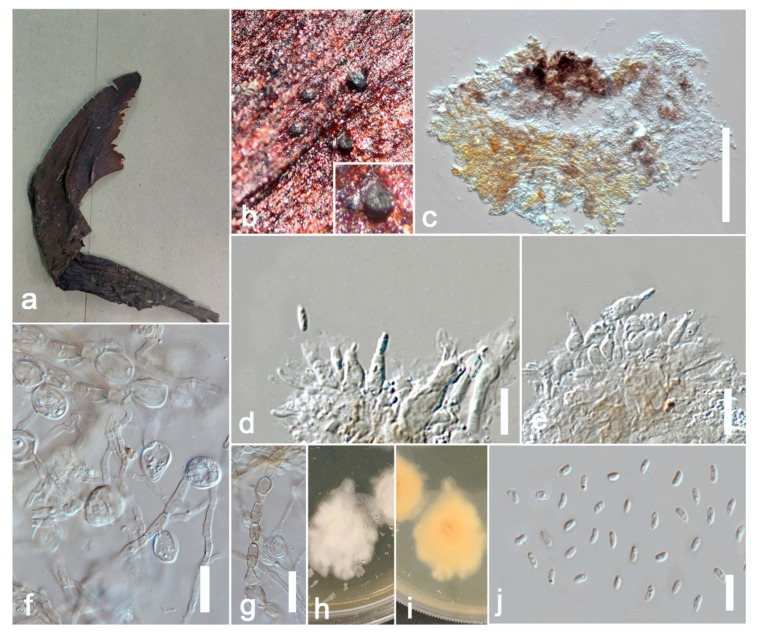
***Allophoma tropica*** (ZHKU 22-0099) (**a**) Examined material. (**b**) Conidiomata on the substrate. (**c**) Vertical section of a conidioma. (**d**,**e**) Conidiogenous cells attached to conidia. (**j**) Conidia. (**f**,**g**) Chlamydospores in cultures. (**h**,**i**) Colonies on PDA (**i** from the bottom). Scale bars: **e** = 200 μm, **f**–**j** = 10 μm.

**Figure 4 jof-09-00182-f004:**
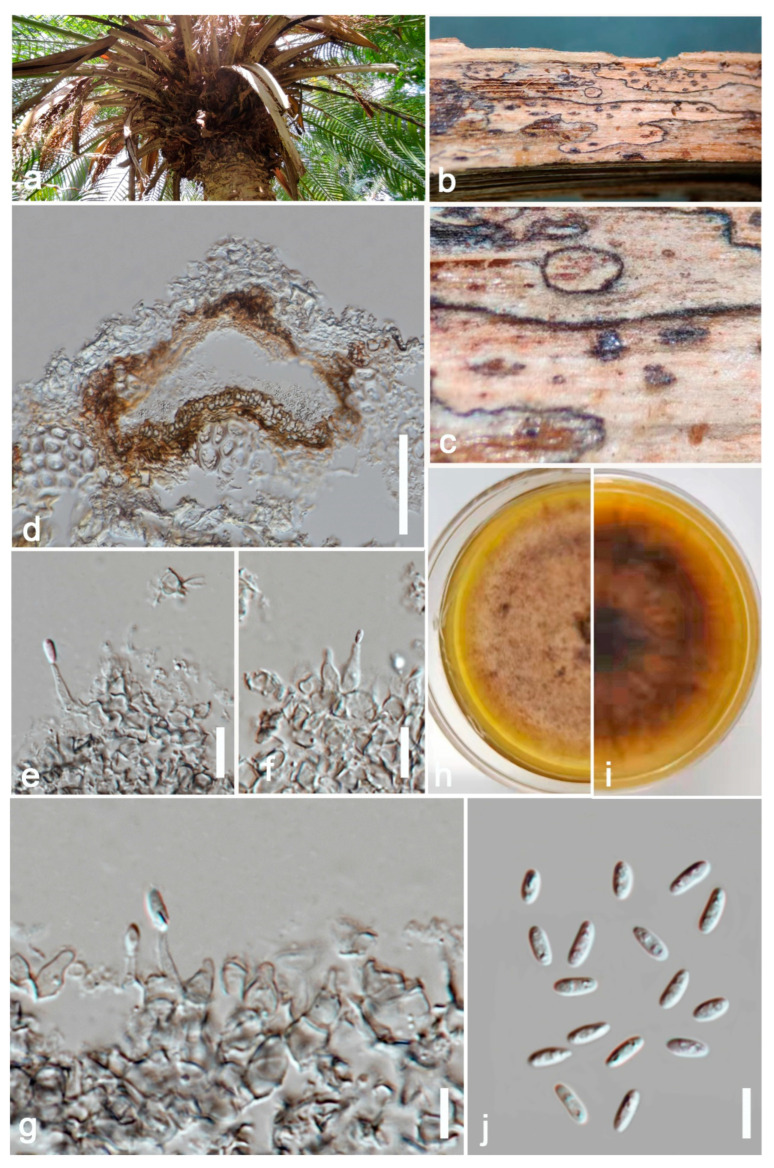
***Ectophoma phoenicis*** (ZHKU 22-0093, **holotype**). (**a**) Host. (**b**,**c**) Conidiomata on the substrate. (**d**) Vertical section of a conidioma. (**e**–**g**) Conidiogenous cells attached to conidia. (**j**) Conidia. (**h**,**i**) Colonies on PDA (**i**) from the bottom. Scale bars: **d** = 100 μm, **e**–**h** = 15 μm.

**Figure 5 jof-09-00182-f005:**
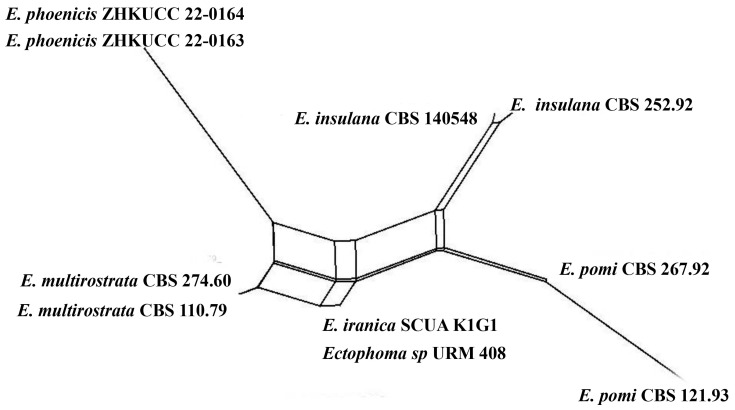
The results of a PHI test of *Ectophoma* clade using both LogDet transformation and splits decomposition. PHI test results Φw = 0.99.

**Figure 6 jof-09-00182-f006:**
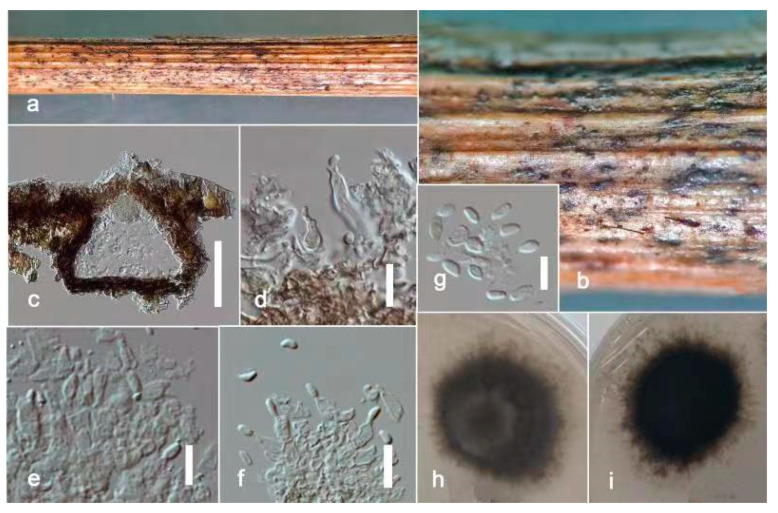
***Remotididymella fici***-***microcarpae*** (ZHKU 22-0095, **holotype**). (**a**,**b**) Conidiomata on the substrate. (**c**) Vertical section of a conidioma. (**d**–**f**) Conidiogenous cells attached to conidia. (**g**) Conidia. (**h**) Surface view of colony on PDA. (**i**) Reverse view of colony on PDA. Scale bars: **c** = 100 μm, **d**–**g** = 15 μm.

**Figure 7 jof-09-00182-f007:**
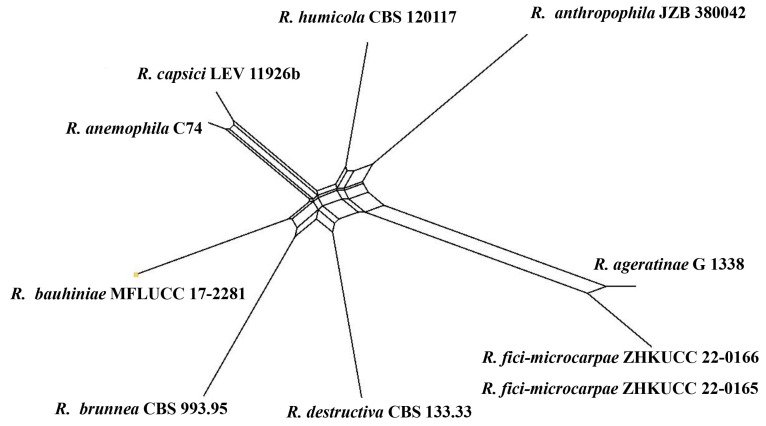
The results of a PHI test of closely related taxa (*Remotididymella* clade) using both LogDet transformation and splits decomposition. PHI test results. Φw = 0.26.

**Figure 8 jof-09-00182-f008:**
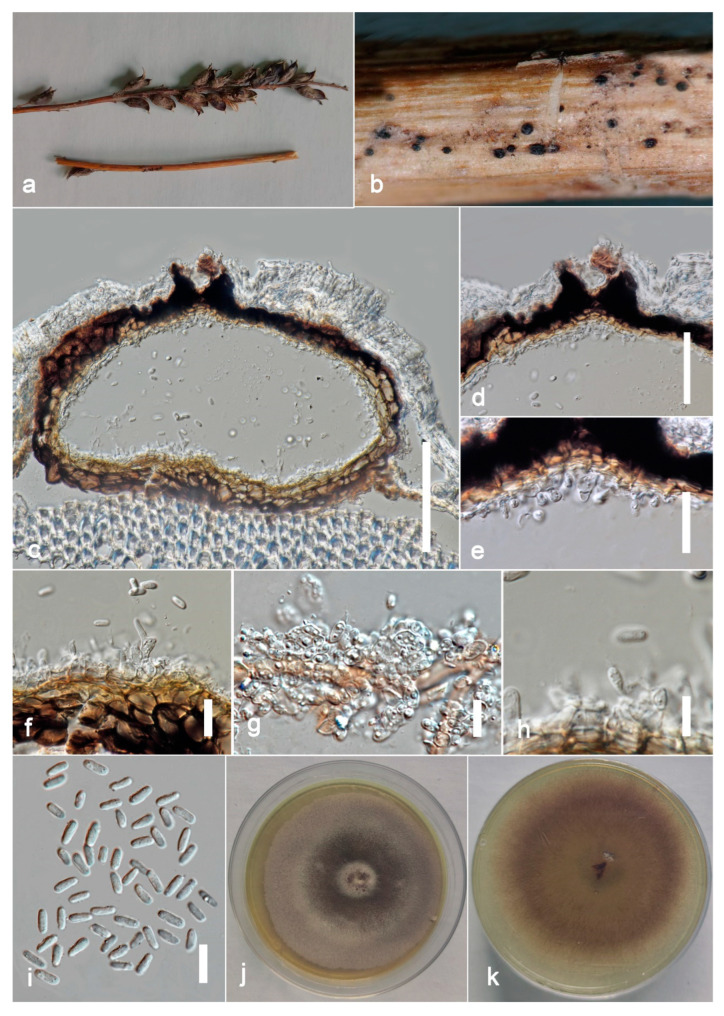
***Stagonosporopsis pedicularis***-***striatae*** (ZHKU 22-0097, **holotype**). (**a**) Host. (**b**) Material examined. (**c**) Vertical section of a conidioma. (**d**) Papilla. (**e**,**f**) Conidiogenous cells attached to conidia. (**g**,**h**) Conidiogenous cells. (**i**) Conidia. (**j**) Colony on PDA (**k**) from the bottom. Scale bars: **c** = 100 μm, **d** = 40 μm, **e**–**i** = 10 μm.

**Figure 9 jof-09-00182-f009:**
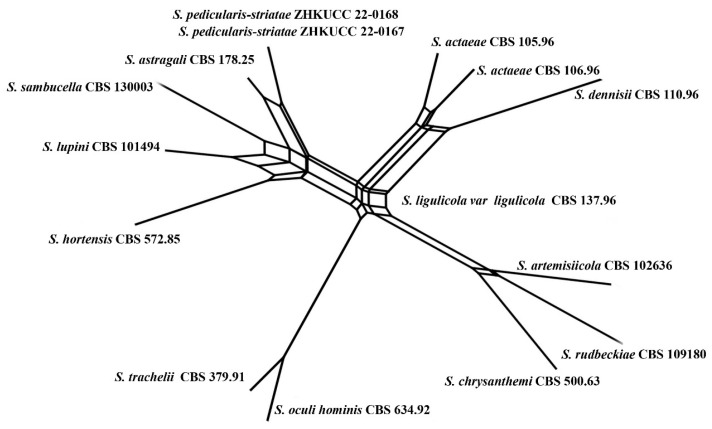
The results of PHI test of closely related taxa (*Stagonosporopsis* sub clade) using both LogDet transformation and splits decomposition. PHI test results. Φw = 0.18.

**Table 1 jof-09-00182-t001:** Gene regions, respective primer pairs, and thermal cycler conditions used in this study.

Gene Region	Primer Pairs	Optimized Thermal Cycler Conditions
ITS	ITS1/ITS4	94 °C: 3 min; (94 °C: 45 s, 56 °C: 50 s) 35 cycles 72 °C: 10 min
LSU	LR0R/LR5	94 °C: 3 min; (94 °C: 45 s, 56 °C: 50 s) 35 cycles 72 °C: 10 min
RPB2	fRPB2-5F/fRPB2-7cR	95 °C: 5 min; (95 °C: 1 min, 52 °C: 50 s) 35 cycles 72 °C: 10 min
*β* *-tubulin*	bt2a/bt2b	95 °C: 5 min; (94 °C: 1 min, 55 °C: 90 s) 35 cycles 72 °C: 10 min

**Table 2 jof-09-00182-t002:** Taxa used in the present phylogenetic analyses and GenBank numbers of their sequences.

Species	Strain no	GenBank Accession Numbers			
		ITS	LSU	RPB2	*β*-*tubulin*
*Allophoma anatiae*	CBS 124673	MN973472	MN943674	MT018048	MT005571
*Allophoma cylindrispora*	UTHSC DI16-233	LT592920	LN907376	LT593058	LT592989
*Allophoma zantedeschiae*	CBS 131.93	FJ427084	GU238159	KT389557	FJ427188
*Allophoma hayatii*	CBS 142859	KY684812	KY684814	MF095108	KY684816
*Allophoma labilis*	CBS 124.93	GU237765	GU238091	KT389552	GU237619
*Allophoma piperis*	CBS 268.93	GU237816	GU238129	KT389554	GU237644
*Allophoma siamensis*	MFLU 17-2422	MK347742	MK347959	MK434912	MK412867
*Allophoma pterospermicola*	LC12182	MK088570	MK088577	MK088584	MK088591
*Allophoma pterospermicola*	CGMCC 3.19245	MK088573	MK088580	MK088587	MK088594
*Allophoma thunbergiae*	GUCC2070.7	MW036298	MW040201	MW116819	MW116823
*Allophoma minor*	CBS 325.82	GU237831	GU238107	KT389553	GU237632
*Allophoma alba*	CBS 120422	MN973469	MN943671	MT018044	MT005568
*Allophoma nicaraguensis*	CBS 506.91	GU237876	GU238058	KT389551	GU237596
*Allophoma tropica*	CBS 436.75	GU237864	GU238149	KT389556	GU237663
*Allophoma tropica*	CBS 537.66	MH858877	MH870533	MT018043	N/A
** *Allophoma tropica* **	**ZHKUCC 22-0169**	**OQ275206**	**OQ275192**	**OQ343373**	**OQ336257**
** *Allophoma tropica* **	**ZHKUCC 22-0170**	**OQ275207**	**OQ275193**	**OQ343374**	**OQ336258**
*Allophoma oligotrophica*	LC 6245	KY742040	KY742194	KY742128	KY742282
*Ectophoma insulana*	CBS 140548	MN973482	MN943686	MT018071	MT005582
*Ectophoma insulana*	CBS 252.92	MN973481	MN943685	MT018070	MT005581
*Ectophoma pomi*	CBS 267.92	GU237814	GU238128	N/A	GU237643
*Ectophoma iranica*	SCUA-K1G1	MK519382	MK519389	N/A	MK519562
*Ectophoma pomi*	CBS 121.93	MN972933	N/A	MN983570	MN983948
*Ectophoma multirostrata*	CBS 110.79	FJ427030	GU238110	LT623264	FJ427140
*Ectophoma multirostrata*	CBS 274.60	FJ427031	GU238111	LT623265	FJ427141
*Ectophoma* sp.	URM 408	MH384820	MH370603	MH370607	N/A
** *Ectophoma phoenicis* **	**ZHKUCC 22-0163**	**OQ275208**	**OQ275194**	**OQ343375**	**OQ336259**
** *Ectophoma phoenicis* **	**ZHKUCC 22-0164**	**OQ275209**	**OQ275195**	**OQ343376**	**OQ336260**
*Remotididymella ageratinae*	G13388	MN864294	MN864298	MN871530	MN871533
*Remotididymella anemophila*	C74	MN864293	MN864296	MN871529	MN871532
*Remotididymella anthropophila*	CBS 142462	LT592936	N/A	LT593075	LT593005
*Remotididymella bauhiniae*	MFLU 18-2118	MK347737	MK347954	MK434914	MK412884
*Remotididymella brunnea*	CBS 993.95	NR_170781	N/A	MT018064	MT005576
*Remotididymella capsici*	CBS 679.77	MN973478	MN943681	MT018066	MT005578
*Remotididymella destructiva*	CBS 133.93	GU237779	GU238064	LT623257	GU237602
***Remotididymella fici*-*microcarpae***	**ZHKUCC 22-0165**	**OQ275210**	**OQ275196**	**OQ343377**	**OQ336261**
***Remotididymella fici*-*microcarpae***	**ZHKUCC 22-0166**	**OQ275211**	**OQ275197**	**OQ343378**	**OQ336262**
*Remotididymella humicola*	CBS 120117	NR_170782	MN943680	MT018065	MT005577
*Stagonosporopsis actaeae*	CBS 106.96	GU237734	GU238166	KT389672	GU237671
*Stagonosporopsis actaeae*	CBS 105.96	GU237733	GU238165	MT018018	GU237670
*Stagonosporopsis ailanthicola*	MFLUCC 16-1439	KY100872	KY100874	KY100876	KY100878
*Stagonosporopsis ajacis*	CBS 176.93	GU237790	GU238167	MT018035	GU237672
*Stagonosporopsis andigena*	CBS 269.80	GU237817	GU238170	MT018026	GU237675
*Stagonosporopsis artemisiicola*	CBS 102636	GU237728	GU238171	KT389674	GU237676
*Stagonosporopsis astragali*	CBS 178.25	GU237792	GU238172	MT018030	GU237677
*Stagonosporopsis bomiensis*	CGMCC 3.18366	KY742123	KY742277	KY742189	KY742365
*Stagonosporopsis chrysanthemi*	CBS 500.63	GU237871	GU238190	MT018012	GU237695
*Stagonosporopsis chrysanthemi*	CBS 137.96	GU237783	GU238191	MT018011	GU237696
*Stagonosporopsis citrulli*	FLAS-F-58996	KJ855546	N/A	N/A	KJ855602
*Stagonosporopsis crystalliniformis*	CBS 713.85	GU237903	GU238178	KT389675	GU237683
*Stagonosporopsis cucumeris*	CBS 386.65	MN973455	MN943657	MT018021	MT005554
*Stagonosporopsis caricae*	CBS 119735	MN973042	N/A	MN983680	MN984054
*Stagonosporopsis caricae*	CBS 102399	MN973041	N/A	MN983679	MN984053
*Stagonosporopsis dennisii*	CBS 110.96	MN973046	N/A	MN983685	MN984058
*Stagonosporopsis dorenboschii*	CBS 320.90	GU237830	GU238184	MT018039	GU237689
*Stagonosporopsis helianthi*	CBS 200.87	KT389545	KT389761	KT389683	KT389848
*Stagonosporopsis hortensis*	CBS 572.85	GU237893	GU238199	KT389681	GU237704
*Stagonosporopsis inoxydabilis*	CBS 425.90	GU237861	GU238188	KT389682	GU237693
*Stagonosporopsis loticola*	CBS 562.81	GU237890	GU238192	KT389684	GU237697
*Stagonosporopsis lupini*	CBS 101494	GU237724	GU238194	KT389685	GU237699
*Stagonosporopsis nemophilae*	CBS 715.85	MN973460	MN943662	MT018031	MT005559
*Stagonosporopsis oculo*-*hominis*	CBS 634.92	GU237901	GU238196	KT389686	GU237701
*Stagonosporopsis papillata*	LC 8171	KY742127	KY742281	KY742193	KY742369
*Stagonosporopsis pini*	MFLUCC 18-1549	MK347800	MK348019	MK434860	MK412886
*Stagonosporopsis rudbeckiae*	CBS 109180	GU237745	GU238197	MT018015	GU237702
*Stagonosporopsis sambucella*	CBS 130003	MN973459	MN943661	MT018029	MT005558
*Stagonosporopsis stuijvenbergii*	CBS 144953	MN823449	MN823300	MN824475	MN824623
*Stagonosporopsis tanaceti*	CBS 131484	NR_111724	KP161044	MT018013	JQ897496
***Stagonosporopsis pedicularis*-*striatae***	**ZHKUCC 22-0167**	**OQ275212**	**OQ275198**	**OQ343379**	**OQ336263**
***Stagonosporopsis pedicularis*-*striatae***	**ZHKUCC 22-0168**	**OQ275213**	**OQ275199**	**OQ343380**	**OQ336264**
*Stagonosporopsis trachelii*	CBS 379.91	GU237850	GU238173	KT389687	GU237678
*Stagonosporopsis valerianellae*	CBS 273.92	GU237819	GU238200	MT018033	GU237705
*Stagonosporopsis weymaniae*	CBS 144959	MN823453	MN823304	MN824479	MN824627
*Stagonosporopsis centaureae*	MFLUCC 16-0787	KX611240	KX611238	N/A	N/A
*Stagonosporopsis heliopsidis*	CBS 109182	GU237747	GU238186	KT389679	GU237691
*Stagonosporopsis pogostemonis*	ZHKUCC 21-0001	MZ156571	MZ191532	MZ203135	MZ203132
*Stagonosporopsis rhizophila*	CGMCC 3.19852	MG833824	MG833789	MN422105	MN422099
*Stagonosporopsis cucurbitacearum*	CBS 133.96	GU237780	GU238181	KT389676	GU237686
*Stagonosporopsis cucurbitacearum*	CBS 109171	GU237922	GU238180	N/A	GU237685
*Stagonosporopsis ailanthicola*	CBS 140554	MN973462	MN943664	MT018036	MT005561
*Epicoccum oryzae*	CBS 173.34	MN973499	N/A	MT018098	MT005599
*Epicoccum viticis*	LC 5126	KY742118	N/A	KY742186	KY742360
*Similiphoma crystallifera*	CBS 193.82	GU237797	GU238060	LT623267	GU237598
*Didymella heteroderae*	CBS 109.92	FJ426983	GU238002	KT389601	FJ427098
*Didymella suiyangensis*	LC 7439	KY742089	KY742243	KY742168	KY742330
*Cumuliphoma indica*	CBS 65478	FJ427044	GU238123	LT623262	FJ427154
*Cumuliphoma pneumoniae*	UTHSC DI16-249	NR_158277	N/A	LT593063	LT592994
*Paraboeremia selaginellae*	CBS 122.93	GU237762	GU238142	MT018189	GU237656
*Paraboeremia litseae*	CGMCC 318109	KX829029	KX829037	KX829045	KX829053
*Macroventuria anomochaeta*	CBS 502.72	GU237873	GU237985	MT018193	GU237545
*Macroventuria wentii*	CBS 526.71	GU237884	GU237986	KT389642	GU237546
*Juxtiphoma eupyrena*	CBS 527.66	FJ427000	GU238073	LT623269	FJ427111
*Juxtiphoma eupyrena*	CBS 374.91	FJ426999	GU238072	LT623268	FJ427110
*Vacuiphoma ferulae*	CBS 353.71	MH860160	MH871928	MT018196	MT005655
*Vacuiphoma oculihominis*	FMR 138.01	LT592954	N/A	LT593093	LT593023
*Nothophoma infossa*	CBS 123395	FJ427025	FJ899743	KT389659	FJ427135
*Nothophoma anigozanthi*	CBS 381.91	GU237852	GU238039	KT389655	GU237580
*Ascochyta medicaginicola*	BRIP 45051	KY742044	KY742198	KY742132	KY742286
*Ascochyta koolunga*	CBS 373.84	KT389481	KT389698	KT389560	KT389775
*Calophoma rosae*	CGMCC 3.18347	KY742049	KY742203	KY742135	KY742291
*Calophoma vodakii*	CBS 173.53	KT389497	KT389714	MT018233	KT389791
*Neomicrosphaeropsis novorossica*	MFLUCC 14-0578	KX198709	KX198710	N/A	N/A
*Neomicrosphaeropsis rossica*	MFLUCC 14-0586	KU752192	KU729855	N/A	N/A
*Phoma herbarum*	CBS 127589	KT389539	MH876049	KT389664	KT389838
*Phoma herbarum*	CBS 134.96	KT389535	KT389753	KT389661	KT389834
*Phomatodes nebulosa*	CBS 740.96	KT389540	KT389758	KT389667	KT389839
*Leptosphaerulina americana*	CBS 213.55	GU237799	GU237981	KT389641	GU237539
*Leptosphaerulina arachidicola*	CBS 275.59	GU237820	GU237983	MT018278	GU237543
*Pseudoascochyta novaezelandiea*	CBS 141689	LT592892	LT592893	LT592895	LT592894
*Pseudoascochyta pratensis*	CBS 141688	LT223130	LT223131	LT223133	LT223132
*Briansuttonomyces eucalypti*	CBS 114879	KU728479	KU728519	MT018239	KU728595
*Xenodidymella applanata*	CBS 115577	KT389546	KT389762	KT389688	KT389850
*Xenodidymella asphodeli*	CBS 375.62	KT389549	KT389765	KT389689	N/A
*Neodidymelliopsis cannabis*	CBS 121.75	GU237761	GU237972	N/A	GU237535
*Neodidymelliopsis achlydis*	CBS 256.77	KT389531	KT389749	MT018293	KT389829
*Neoascochyta europaea*	CBS 819.84	KT389510	KT389728	KT389645	KT389808
*Neoascochyta graminicola*	CBS 102789	KT389518	KT389736	KT389649	KT389816
*Leptosphaeria conoidea*	CBS 616.75	JF740201	JF740279	KT389639	KT389804
*Heterophoma poolensis*	CBS 113.20	GU237751	GU238119	MT018056	GU237638
*Boeremia diversispora*	CBS 101194	GU237716	GU237929	KT389564	GU237491
*Boeremia foveata*	CBS 109176	GU237742	GU237946	KT389578	GU237508
*Leptosphaeria doliolum*	CBS 505.75	JF740205	GU301827	KT389640	JF740144
*Heterophoma nobills*	CBS 507.91	GU237877	GU238065	KT389638	GU237603

**Abbreviations**: **CBS**, Westerdijk Fungal Biodiversity Institute, Utrecht, The Netherlands; **CGMCC**, China General Microbiological Culture Collection, Beijing, China; **MFLUCC**, Mae Fah Luang University Culture Collection, Chiang Rai, Thailand; **BMU**: Department of Dermatology, Beijing Medical University, Beijing, China; **UTHSC**, Fungus Testing Laboratory at the University of Texas Health Science Center, San Antonio, Texas, USA; **BRIP**, Plant Pathology Herbarium, Department of Employment, Economic, Development and Innovation, Queensland, Australia; **FMR**, Facultat de Medicina, Universitat Rovira i Virgili, Reus, Spain; **LC**, Cai’s personal collection deposited in laboratory, housed at CAS, PR China; **SCUA**: the Collection of Fungal Cultures, Department of Plant Protection, Shahid Chamran University of Ahvaz, Iran; **GZCC**, Guizhou culture collection, Guizhou, China; **ZHKUCC**, University of Agriculture and Engineering Culture Collection (China). “N/A” denotes sequences that are not available. Newly generated sequences in this study are indicated in bold.

## Data Availability

All sequence data are available in NCBI GenBank following the accession numbers in the manuscript.

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
