# Peer review of "Plant-Associated Novel Didymellaceous Taxa in the South China Botanical Garden (Guangzhou, China)"

_jof, 2023, doi:10.3390/jof9020182_

Round 1

Reviewer 1 Report

Suggestions and corrections are found in the attachement.

Author Response

Dear Sri /Madam.

Thank you for your valuable suggestion on my manuscript 

Plant-associated novel didymellaceous taxa in the south China Botanical Garden (Guangzhou, China).

I corrected all the suggestions you made on My MS.

Response to Reviewer 1 Comments

Point 1: Currently, the family encompasses more than 5,400 species in more than 31 accepted genera [12–14] Line 42

Response 1: It was change into 44 genera acording to the outline of fungi 2022.

Point 2: They are mostly plant pathogens Line 44-49

Response 2: It was change into some of are plant pathogens.

Point 3: author abbreviations should be used. Missing for: Allophoma tropica Line 57

Response 3: It was corrected to Allophoma tropica (R. Schneid. & Boerema) Qian Chen & L. Cai.

Point 4: Missing dot behind: Senanayake et al Line 73

Response 4: It was corrected adding dot after Senanayake et al.

Point 5: correct size of: the RNA polymerase II subunit (rpb2) gene and Line 92

Response 5: It was change into normal font size.

Point 6: Author abbreviations are missing for: Leptosphaeria conoidea and Leptosphaeria
doliolum
Line 111-112

Response 6         : It was corrected to Leptosphaeria conoidea (De Not.) Sacc. and Leptosphaeria doliolum (Pers.) Ces. & De Not.

Point 7: Author abbreviations are missing for Stagonosporopsis astragali Line 152

Response 7: It was change into Stagonosporopsis astragali (Cooke & Harkn.) Aveskamp, Gruyter & Verkley.

Point 8: Author abbreviations are missing for Remotididymella ageratinae Line 159

Response 8: It was change into Remotididymella ageratinae H.B. Zhang, A.L. Yang & L. Chen

.

Point 9: Newly generated sequences should be marked in bold in the Table 2 as indicated by the
captions below the table Line 161

Response 9: It was marked and bold in the table.

Point 10: Author abbreviations are missing for Allophoma labilis Line 192

Response 10: It was change into Allophoma labilis (Sacc.) Qian Chen & L. Cai.

Point 11: Author abbreviations are missing for Allo. pterospermicola and genus should be used as A. Line 232

Response 11: It was change into A. pterospermicola

Response 11: It was change into A. pterospermicola Qian Chen & L. Cai

Point 12: missing for: Allo. thunbergiae and use A. thunbergiae Line 233

Response12: It was change into A. thunbergiae.

Response12: It was change into A. thunbergiae Jun Yuan & Yong Wang.

Point 13: Change italics style in Type species for regular style Line 241

Response 13: It was change regular style.

Point 14: Author abbreviations are missing for: P. pereupyrena Line 242

Response 14: It was change into P. pereupyrena Gruyter, Noordel. & Boerema.

Point 15: Author abbreviations are missing for also for: E. iranica, E. multirostrata; Remove highlights of percentages % Line 274

Response 15: It was change into Ectophoma multirostrata (P.N. Mathur, S.K. Menon & Thirum.) Valenz.-Lopez, Cano, Crous, Guarro & Stchigel  and E. Iranica M. Mehrabi, Larki & Farokhinejad.

Response 15: Highlights % was removed.

Point 16: Behind first author abbreviation is duplicity: Kular., & Use only comma Line 251

Response 16: It was changed.

Point 17: Etymology in italics change style for regular Line 253

Response 17: It was change into regular font.

Point 18: Add author abbreviations R. bauhiniae Line 308

Response 18: It was change into R. bauhiniae Jayasiri, E.B.G. Jones & K.D. Hyde

Point 19: For Remotididymella fici-microcarpae use bold style Line 314

Response 19: It was change into bold.

.

Point 20: At the end of the sentence use . (dot) Line 370

Response 20: It was inserted.

Point 21: Author abbreviations are missing for Stagonosporopsis vannaccii; Line 451

Response 21: It was change into Stagonosporopsis vannaccii Baroncelli, Cafà, Castro, Boufleur & Massola.

Point 22: Author abbreviations are missing for S. cucurbitacearum Line 452

Response 22: It was change into Stagonosporopsis cucurbitacearum (Fr.) Aveskamp, Gruyter & Verkley.

Point 23: Author abbreviations are missing for Stagonosporopsis pogostemonis Line 454

Response 23: It was change into M. Luo, Y.H. Huang & Manawas.

Point 1: Use correct font and size for: 99  Line 511

Response 24: It was change into body font size.

Point 2: without italics: sp. nov Line 530

Response 25: It was change into regular font.

Point 3: delete dot: Dermatology. 1981 Line 531

Response 26: It was corrected.

Point 4: Use correct font and size for: 470.  Line 596

Response 27: It was corrected.

Point 5: reference no. 65 is not on correct line and in correct style Line 629

Response 28: It was change into correct reference number.

Best regards 

Nuwan

Reviewer 2 Report

The paper by Kularathnage et al., is a good piece of work, well written, easy to read; the experimental plan and methods are appropriate and up to date. My only concern is on the sampling procedure that should be better described: what are the materials sampled? infected plant material? which part? stems? leaves? how have been collected? what sterile procedure and how they brought the materials to the pab and preserved them until being analysed? Few further comments and corrections are along the text attached

Author Response

Dear Sri /Madam.

Thank you for your valuable suggestion on my manuscript 

Plant-associated novel didymellaceous taxa in the south China Botanical Garden (Guangzhou, China).

I corrected all the suggestions you made on My MS.

Point 1: What specimens? Infected plant material? Please specify. What was the exact sampling procedure? sterile sampling? specimens preserved in sterile bags? Line 62

Response 1: It was change into Specimens were detached from the host using sterile blades and packed in sterilized paper bags.

Point 2: What samples (infected plant material? Stems, leaves? and how did you carried them to the lab? sterile sampling? sterile procedure Line 69

Response 2: It was change into The collected dead samples (petioles, sepals and stems) were brought to the laboratory in sterilized paper bags.

Point 3: Font italica Line 251

Response 3: It was changed into italics.

Point 4: scale bar Line352

Response 4: It alredy mention starting from d and end with g.

Point 5: mediaThere

Response 5: It was change into media. There adding fulstop.

Point 6: correct numbering Line 626

Response 6         : It was corrected removing number 1.

Best regards 

Nuwan